# COVID-19 Vaccine Perceptions among Ebola-Affected Communities in North Kivu, Democratic Republic of the Congo, 2021

**DOI:** 10.3390/vaccines11050973

**Published:** 2023-05-11

**Authors:** Stephanie Chow Garbern, Shiromi M. Perera, Eta Ngole Mbong, Shibani Kulkarni, Monica K. Fleming, Arsene Baleke Ombeni, Rigobert Fraterne Muhayangabo, Dieula Delissaint Tchoualeu, Ruth Kallay, Elizabeth Song, Jasmine Powell, Monique Gainey, Bailey Glenn, Hongjiang Gao, Ruffin Mitume Mutumwa, Stephane Hans Bateyi Mustafa, Neetu Abad, Gnakub Norbert Soke, Dimitri Prybylski, Reena H. Doshi, Rena Fukunaga, Adam C. Levine

**Affiliations:** 1Department of Emergency Medicine, Alpert Medical School of Brown University, Providence, RI 02903, USA; 2International Medical Corps, Washington, DC 20036, USA; 3International Medical Corps, Goma, Democratic Republic of the Congo; 4Global Immunization Division, Centers for Disease Control and Prevention, Atlanta, GA 30333, USA; 5Brown University, Providence, RI 02912, USA; 6Rhode Island Hospital, Providence, RI 02903, USA; 7James A. Ferguson Infectious Disease Program, Centers for Disease Control and Prevention, Atlanta, GA 30333, USA; 8Expanded Programme on Immunization, Goma, Democratic Republic of the Congo; 9Division of Global Health Protection, Centers for Disease Control and Prevention, Kinshasa, Democratic Republic of the Congo; 10Division of Global HIV and TB, Centers for Disease Control and Prevention, Atlanta, GA 30333, USA

**Keywords:** SARS-CoV-2, Ebola Virus Disease, pandemic, outbreak, Democratic Republic of the Congo, humanitarian, vaccines, vaccine acceptance, vaccine hesitancy

## Abstract

Populations affected by humanitarian crises and emerging infectious disease outbreaks may have unique concerns and experiences that influence their perceptions toward vaccines. In March 2021, we conducted a survey to examine the perceptions toward COVID-19 vaccines and identify the factors associated with vaccine intention among 631 community members (CMs) and 438 healthcare workers (HCWs) affected by the 2018–2020 Ebola Virus Disease outbreak in North Kivu, Democratic Republic of the Congo. A multivariable logistic regression was used to identify correlates of vaccine intention. Most HCWs (81.7%) and 53.6% of CMs felt at risk of contracting COVID-19; however, vaccine intention was low (27.6% CMs; 39.7% HCWs). In both groups, the perceived risk of contracting COVID-19, general vaccine confidence, and male sex were associated with the intention to get vaccinated, with security concerns preventing vaccine access being negatively associated. Among CMs, getting the Ebola vaccine was associated with the intention to get vaccinated (RR 1.43, 95% CI 1.05–1.94). Among HCWs, concerns about new vaccines’ safety and side effects (OR 0.72, 95% CI 0.57–0.91), religion’s influence on health decisions (OR 0.45, 95% CI 0.34–0.61), security concerns (OR 0.52, 95% CI 0.37–0.74), and governmental distrust (OR 0.50, 95% CI 0.35–0.70) were negatively associated with vaccine perceptions. Enhanced community engagement and communication that address this population’s concerns could help improve vaccine perceptions and vaccination decisions. These findings could facilitate the success of vaccine campaigns in North Kivu and similar settings.

## 1. Introduction

The COVID-19 vaccination is considered the most effective way to reduce morbidity and mortality from COVID-19; however, large disparities in vaccine access and acceptance persist globally [1,2]. As of December 2022, only 23% of the population in the African Region completed the primary COVID-19 vaccine series—far from the World Health Organization’s (WHO) global target of 70% by the end of 2022 [3]. There are multiple reasons for this slow progress including operational challenges, such as vaccine storage and distribution issues, scarce local vaccine manufacturing capacities, and low vaccine demand due to a lack of perceived risk from COVID-19 in Africa [4,5].

The Democratic Republic of the Congo (DRC) has one of the lowest COVID-19 vaccination rates in the world, having fully vaccinated 12% of its 100 million estimated population as of April 2023 [6]. Vaccinations in the DRC began with the rollout of the ChAdOx1-S (recombinant) (AstraZeneca^®^/Covishield) vaccine in April 2021 and subsequently in September 2021 with the Moderna mRNA-1273 and Pfizer-BioNTech (BNT162b2) COVID-19 vaccines, through the COVID-19 Vaccines Global Access (COVAX) facility, and the Sinovac vaccine through a bilateral agreement between the DRC and China [6,7,8,9]. Specific challenges to COVID-19 vaccine uptake in the DRC include healthcare worker (HCW) shortages and strikes, concurrent outbreaks of other vaccine-preventable diseases (VPDs), including Ebola Virus Disease (EVD), and high rates of vaccine hesitancy due to misinformation [7,10]. A 2020 survey conducted by the Africa Centers for Disease Control and Prevention (Africa CDC) found that DRC respondents had the lowest willingness to receive a vaccine among the 15 African countries included in that survey (59% DRC vs. >90% in Ethiopia, Nigeria, and Tunisia) [11]. Significant regional variations in COVID-19 vaccine attitudes exist, with the willingness to vaccinate ranging from <40% in Kwilu to >90% in Kasaï-Central in a 2020 survey [12]. Additionally, previous studies have discovered high mistrust toward COVID-19 vaccines among HCWs in the DRC. One study revealed that only 27.7% of HCWs across the 23 referral hospitals were willing to receive the COVID-19 vaccine [13].

North Kivu, a province located in the Eastern DRC, has been the site of decades-long humanitarian emergencies and an active conflict zone. It has also been the site of multiple outbreaks of EVD (including the most recent in August 2022) and was the epicenter of the 2018–2020 EVD outbreak, which was the second largest in history, resulting in over 3400 cases and 2200 deaths [14]. Low vaccination rates for VPDs including COVID-19 have been observed in populations affected by humanitarian crises due to healthcare infrastructure and societal disruptions, political instability, heightened mistrust toward health authorities, as well as security concerns, and violence against healthcare facilities and workers [15,16,17,18]. North Kivu communities affected by the 2018–2020 EVD outbreak are unique given their prior experience with another novel vaccine, the Ebola (ERVEBO rVSV∆G-ZEBOV-GP) vaccine, which was deployed under an Expanded Access/Compassionate Use ring vaccination protocol [19]. Despite the Ebola vaccine’s proven efficacy, it was initially met with resistance. Research pointed to complex drivers of Ebola vaccine hesitancy including skepticism concerning vaccine efficacy and necessity, government mistrust, and beliefs about foreign organizations harboring ulterior motives [20,21,22,23]. Ultimately, the Ebola vaccine acceptance increased to over 90% of those eligible receiving the vaccine during the outbreak; this was due to community engagement, responsiveness to concerns about vaccine eligibility, and increasing trust [24,25,26]. 

Recent data show that COVID-19 vaccine acceptance in North Kivu has been very low since the vaccine rollout began. North Kivu ranked in the bottom five of all 25 DRC provinces for vaccination coverage [27]. As of February 2023, less than 5% of the eligible population in North Kivu has been fully vaccinated compared to 10% in Kinshasa and over 50% in Kasai Oriental [28]. North Kivu’s historical experiences (e.g., protracted multi-faceted conflict and repeated Ebola outbreaks with mandatory vaccination for HCWs) may now shape how these communities respond to developing global health threats (e.g., COVID-19 and willingness to receive COVID-19 vaccines). Identifying the reasons behind vaccination decisions could inform continued efforts to reduce COVID-19 vaccine hesitancy in this population and others that are similarly exposed to emerging infectious diseases (EID) and conflict. This survey aimed to examine the perceptions toward COVID-19 vaccines and identify the factors associated with the intention to receive the COVID-19 vaccine among community members (CMs) and HCWs affected by the 2018–2020 North Kivu EVD outbreak, which was shortly before the COVID-19 vaccine rollout in the DRC in April 2021.

## 2. Materials and Methods

### 2.1. Survey Design, Population, and Setting

We collected data on perceptions toward COVID-19 and COVID-19 vaccines as part of a larger cross-sectional survey on Ebola vaccine acceptance in North Kivu, DRC. Surveys were administered from 5 March to 16 March 2021 in three health zones (Beni, Mabalako, and Butembo) that had active EVD transmission during the 2018–2020 EVD outbreak (Appendix A). Detailed methods of the parent survey are described elsewhere [29]. At the time of data collection, COVID-19 vaccines were not yet available and vaccine introduction officially began in May 2021 in North Kivu.

Survey participants were recruited in two categories: (1) high-risk CMs, including EVD survivors, their household members, and members of survivors’ neighboring households who were likely offered the Ebola vaccine (Merck ERVEBO rVSV∆G-ZEBOV-GP) as part of the ring vaccination strategy and (2) HCWs, including any personnel who had been working in a health facility (e.g., hospital or Ebola treatment center) during the 2018–2020 EVD outbreak. Only consenting adults aged 18 years or older were eligible for participation. Participation was anonymous, voluntary, and uncompensated.

#### 2.1.1. Selection of CMs

A modified cluster sampling strategy mimicking ring vaccination was used to enroll individuals who were most likely to be eligible for the Ebola vaccine. First, we obtained a list of Ebola survivors from the survivors’ association (a voluntary community organization of Ebola survivors). Then, we randomly selected 39 survivors (each with approximately ten contacts, to reach the target sample size of the parent survey) from this list. All adult members of the selected survivors’ households were approached for enrollment. In addition, a spin-the-bottle technique was used to select the closest neighboring household to the survivor’s household and enroll adult members of that household as well. The process of selecting the closet neighboring household was repeated until at least ten adult participants were enrolled in each survivor cluster [30].

#### 2.1.2. Selection of HCWs

There were 187 eligible health facilities (81 in Butembo, 79 in Beni, and 27 in Mabalako) that were functional during the time of the 2018–2020 Ebola outbreak. We chose a random sample of 100 health facilities from these three selected health zones. The sampling frame included small (e.g., health posts and dispensaries) and large facilities (e.g., hospitals and referral centers), and public and private facilities. All HCWs in each selected health facility had the opportunity to complete the survey.

### 2.2. Survey Development and Implementation

Questions related to COVID-19 vaccines were added to the parent questionnaire because of the ongoing COVID-19 pandemic at the time of the survey. The structured questionnaire included information on the respondent’s demographics, knowledge, and perceptions about COVID-19 and COVID-19 vaccines, intention to receive the COVID-19 vaccine, vaccine communication preferences, and general vaccine confidence (i.e., perceptions toward routine immunizations). We explained the purpose of the survey to all participants and gave additional details on an information sheet in the local language. Due to low literacy rates and the need to limit physical contact during the COVID-19 pandemic, trained research staff obtained verbal informed consent and documented it electronically on the data collection tool.

### 2.3. Data Collection 

Trained multi-lingual interviewers who were fluent in the local languages (French and Swahili) worked in pairs (male and female) and recorded responses on mobile devices preprogrammed with KoBoCollect [31]. They adhered to social distancing precautions and used appropriate personal protective equipment (PPE). Participation was anonymous, voluntary, and uncompensated. 

### 2.4. Outcome and Explanatory Variables

The outcome variable for the regression analysis was defined as the intention to receive the COVID-19 vaccine based on the participants’ response to the question, “Will you take a COVID-19 vaccine if offered?”. Responses of “I will take a COVID-19 vaccine” were coded as positive and responses of “I will not take a COVID-19 vaccine” and “I am not sure if I will take a COVID-19 vaccine” were coded as negative (i.e., “vaccine-hesitant”).

Explanatory variables were selected for inclusion in the regression model based on expert consensus, Health Beliefs Model (HBM), and literature review to create a working model that illustrates the main factors influencing vaccine hesitancy and acceptance of novel vaccines [32,33]. Sociodemographic variables (sex, age, and education level) were included a priori as explanatory variables.

Variables included in the regression model were sex, age, highest education level attained, the influence of religion, perceived risk of contracting COVID-19, the perception of vaccine side effects as important, the perception of vaccine efficacy as important, the belief that new vaccines pose more risks (i.e., are less safe), distrust toward government to make vaccine decisions, and security concerns preventing access to vaccine and health services. Ebola vaccination status was included in the regression model for CMs, although it was not included for HCWs because nearly all eligible HCWs had received the Ebola vaccine. 

A general vaccine confidence composite score, based on a test that had been previously validated in Sierra Leone, was calculated using a segment of six questions designed to assess perceptions toward routine immunizations [34]. The total score ranged from 0 to 18, corresponding to low–high vaccine acceptance. Cronbach’s alpha for the scale was 0.79, indicating high internal reliability. The composite score was then categorized as low, medium, or high vaccine acceptance corresponding to a total score of <6 (low), 6–12 (medium), and >12 (high). Lastly, a COVID-19 knowledge score was created by tallying the responses from a subset of questions on COVID-19 knowledge, awareness, and perception. For each positive response, one point was given for a total COVID-19-related knowledge score, ranging from 0 to 7, which was then categorized as “low” (score 0–2), “medium” (score 3–5), and “high” (score > 5).

### 2.5. Data Analysis 

Descriptive analyses using frequencies with percentages, medians with interquartile ranges (IQR), or means with standard deviations (SD) were performed as appropriate. Stata Version 16 (Stata Corp; College Station, TX, USA) was used for all data analyses. We used a modified Poisson regression model to analyze the data for community members. Modified Poisson regression is an alternative to binomial logistic regression and is used to estimate risk ratios for binary response variables from clustered prospective data when the outcome is not rare [35,36]. Using Stata’s xtgee procedure, we assessed the associations between independent explanatory variables and the outcome of COVID-19 vaccine acceptance, with clustering specified at the level of each survivor cluster (i.e., a cluster including EVD survivors, their household members, and the members of neighboring households) [35,36]. To analyze the HCW data, Stata’s svyset procedure was used to specify clustering at the health facility level, applying sampling weights to account for the cluster survey design. Binary logistic regression was used for the HCW multivariable analysis, yielding adjusted odds ratios. Stratifications of the health facilities as primary or secondary facilities were included in the survey sampling design. Goodness-of-fit was assessed for the regression models using the F-adjusted mean residual test for the HCW data and chi-square goodness-of-fit test for the community members’ data [37].

### 2.6. Ethical Considerations

The University of Kinshasa School of Public Health Ethics Committee approved the survey (protocol approval #203-2020). The assessment was reviewed by the U.S. Centers for Disease Control and Prevention (CDC) and it was determined to be a non-research public health activity. 

## 3. Results

### 3.1. Respondent Characteristics

A total of 631 CMs and 438 HCWs met the inclusion criteria and consented to participate in the survey (Appendix A). The median (IQR) age of CMs was 31 (22–42) years (range 18–88); the IQR age for HCWs was 35 (29–42) years (range 18–75) (Table 1). Females represented 67% of the CMs and 53.7% of the HCWs. More than half of the CMs (60.2%) and nearly all the HCWs (89.7%) had at least a secondary school education (Table 1). Three-quarters (75.1%) of the CMs were offered the Ebola vaccination during the 2018–2020 EVD outbreak, and of those, 83.8% received the vaccine. Nearly all HCWs were offered the Ebola vaccination (95.9%), and nearly all of those offered received the vaccine (99.0%) (Table 1).

### 3.2. COVID-19 Knowledge and Awareness

Knowledge and awareness regarding COVID-19 were high among both respondent groups. Nearly all CMs (97.9%) and HCWs (99.1%) reported awareness of COVID-19 (Table 2). Slightly over half (53.6%) of CMs but most (81.7%) HCWs felt at risk of contracting COVID-19. Most respondents reported that they knew COVID-19 could be spread person-to-person (62.1% of the CMs; 69.4% of the HCWs). They also knew that wearing a mask (85.6% of the CMs; 95.9% of the HCWs) and washing one’s hands regularly (78% of the CMs; 83.6% of the HCWs) could help prevent COVID-19 (Table 2). The median (IQR) score for COVID-19 knowledge was 4 (3–5)among CMs and 5 (4–6) among HCWs.

### 3.3. COVID-19 Vaccine Perceptions 

Perceptions toward COVID-19 vaccines were mixed among both the CMs and HCWs. Roughly half of both respondent groups felt that COVID-19 vaccines were an important measure to control COVID-19; 21.6% of the CMs and 23.1% of the HCWs strongly agreed and 16.3% of the CMs and 25.1% of the HCWs agreed that the COVID-19 vaccine was necessary to stop the spread of the disease, while the remainder were unsure or disagreed (Appendix A). Both groups perceived other prevention measures to be of high importance regardless of vaccination; only 3.0% of the CMs and 1.4% of the HCWs strongly agreed that once vaccinated, other prevention measures would be unnecessary (Appendix A). Lastly, respondents most frequently strongly agreed (27.6% CMs; 34.7% HCWs) that everyone should be offered the COVID-19 vaccine (i.e., not just those at increased risk) (Appendix A). The responses to other Likert scale questions on perceptions toward COVID-19 vaccines are shown in Appendix A.

### 3.4. Intention to Receive the COVID-19 Vaccine 

The intention to receive the COVID-19 vaccine was low in both groups; only 27.6% of the CMs and 39.7% of the HCWs reported that they would receive a vaccine if offered. Most respondents were classified as vaccine-hesitant, with 22.5% of the CMs and 26.5% of the HCWs being unsure if they would receive the COVID-19 vaccine, and 48.2% of the CMs and 32.4% of the HCWs reporting that they would not receive the COVID-19 vaccine (Table 3). Among the respondents who would receive a vaccine or were unsure, the most common reason for wanting the COVID-19 vaccine was to protect themselves and their families (CMs 76.6%; HCWs 74.5%). Not having enough information about the vaccine (53.0% CMs; 66.2% HCWs) and concerns about the vaccine’s safety (37.2% of the CMs; 36.6% of the HCWs) were the most common reasons for vaccine refusal among those who did not want a vaccine or were unsure (Table 3). Lastly, more than half (54.6%) of the HCW respondents would encourage patients to receive the COVID-19 vaccine, while 20.5% would not and 22.6% were unsure (Table 3). The additional reasons provided by the respondents regarding whether they would receive or refuse a vaccine are shown in Table 3. 

### 3.5. General Vaccine Confidence and Vaccine Decision-Making

General vaccine confidence (i.e., perceptions toward routine immunizations) among all respondents was high. Among the CMs, 72.9% agreed (“very much” or “somewhat”) that vaccines were good, and 77.3% agreed that vaccines protect against diseases (Appendix A). Among the HCWs, 88.6% agreed that vaccines were good, and 87.9% agreed that vaccines protect against diseases (Appendix A). The median (IQR) vaccine confidence composite scores were 12 (9–15)and 14 (11–16) among CMs and HCWs, respectively (Appendix A). Regarding vaccine decision-making, 404 (64%) CMs and 318 (72.6%) HCWs reported that information regarding vaccine efficacy was important. Additionally, 285 (45.2%) CMs and 209 (47.7%) HCWs reported that information about vaccine side effects was critical (Appendix A).

### 3.6. Vaccine-Related Communication Preferences

Both CMs and HCWs stated that the most preferred methods for receiving vaccine- and health-related communications were from the radio (73.1% CMs; 80.6% HCWs), HCWs (56.4% CMs; 71.5% HCWs), religious venues (49.3% CMs; 61.9% HCWs), and megaphone announcements (42.6% CMs; 43.8% HCWs), as shown in Table 4. The HCWs stated that the best communication methods for sharing vaccine information among HCWs were meetings or workshops (85.6%), posters (51.8%), pamphlets or handouts (42.0%), and memory aids (24.7%).

### 3.7. Correlates of COVID-19 Vaccine Acceptance

We included only 618 CMs and 432 HCW respondents in our final analysis with the regression model because 13 CMs and 6 HCWs declined to answer whether they intended to receive the COVID-19 vaccine. Among the CMs, the intention to receive the COVID-19 vaccine was associated with the perceived risk of contracting COVID-19 (adjusted RR (aRR) 2.17; 95% CI [1.54, 3.07]), high general vaccine confidence (aRR 3.01; 95% CI [1.51, 5.99]), and prior receipt of the Ebola vaccine (aRR 1.43; 95% CI [1.05, 1.94]). Factors that were negatively associated with vaccine acceptance included the female sex (aRR 0.78; 95% CI [0.63, 0.96]), the perception that new vaccines pose more risks (aRR 0.63; 95% [CI 0.45, 0.86]), and security concerns preventing access to vaccinations and health services (aRR 0.70; 95% CI [0.51, 0.96]) (Table 5). The model’s F-adjusted mean residual test *p*-value was 0.31, indicating no evidence of lack of fit.

Among the HCWs, the intention to receive the COVID-19 vaccine was significantly correlated with the perceived risk of contracting COVID-19 (adjusted OR (aOR) 2.23; 95% CI [1.53, 3.26]) and high general vaccine confidence (aOR 7.16; 95% CI [3.89, 3.17]). The factors that were negatively associated with the intention to get vaccinated included the following: the female sex (aOR 0.60; 95% CI [0.46, 0.80]), religion influencing some health decisions (aOR 0.45; 95% CI [0.34, 0.61]), the belief that new vaccines pose more risks (aOR 0.23; 95% CI [0.17, 0.33]), the perception that vaccine side effects are important (aOR 0.72; 95% CI [0.57, 0.91]), distrust toward the government on vaccine decisions (aOR 0.50; 95% CI [0.35, 0.70]), and security concerns preventing access to vaccinations and health services (aOR 0.52; 95% CI [0.37, 0.74]) (Table 6). The model’s chi-square goodness-of-fit test *p*-value was 0.18 indicating no evidence of lack of fit.

## 4. Discussion

To our knowledge, this is the first survey that specifically explores perceptions toward COVID-19 vaccines and intentions to receive vaccine among a sub-population that had previous firsthand experience with the Ebola vaccine. We found that the intention to receive the COVID-19 vaccine was low (less than one-third of CMs and less than 40% of HCWs) in North Kivu about a month before the COVID-19 vaccine introduction. These findings are consistent with previous research demonstrating that the DRC, particularly the North Kivu province, had one of the lowest rates of willingness to receive the COVID-19 vaccine in Africa during the pre-rollout period [11,12]. 

It is important to note that this survey was implemented between March 5 and 16, 2021, overlapping with the arrival of the DRC’s first shipment of 1.8 million vaccine doses from the COVAX facility [38,39,40]. However, on March 15, the vaccine campaign was paused due to widely publicized concerns about the rare but serious thrombotic side effects of the Oxford–AstraZeneca vaccine [7,40]. Due to low acceptance and the realization that most doses would expire, the DRC returned over 1.3 million vaccine doses to COVAX. Vaccination subsequently began in Kinshasa on 19 April, and then in North Kivu in early May. Owing to slow acceptance, fewer than 6000 doses were administered in North Kivu by June 2021 [7,41]. Unfortunately, vaccine acceptance has remained extremely low, with an actual vaccine acceptance level of <5% in North Kivu as of the end of 2022—even lower than the rates of intention to vaccinate found in this survey [28]. This discrepancy between intention and actual acceptance suggests that additional barriers persist even for those with favorable perceptions toward vaccines in the pre-rollout period [28]. 

Despite the high knowledge and awareness of COVID-19, the perceived risk of COVID-19 varied, with most HCWs feeling at risk of contracting COVID-19 versus only half of the CMs. Among both groups, high general vaccine confidence and perceived risk of contracting COVID-19 were associated with willingness to receive the COVID-19 vaccine, while females and those who reported security concerns preventing vaccine access were less willing to be vaccinated. CMs who received an Ebola vaccine during the 2018–2020 outbreak were also more willing to receive the COVID-19 vaccine than other CMs. The lack of information and concerns about vaccine safety were the most common reasons for unwillingness to receive the COVID-19 vaccine. Among the HCWs, those who believed the new vaccines were riskier, had concerns about vaccine side effects, and who distrusted the government for vaccine decisions were less willing to receive the COVID-19 vaccine. 

Recent data from an intensive vaccine communication campaign in North Kivu, conducted by the non-governmental organization SANRU, have found higher vaccination rates (1386/2350; 59% in Beni) among those who received targeted sensitization via health workers, public meetings, and radio (unpublished data), which is consistent with the preferred communication methods found in this survey. Transparent messaging that emphasizes the risks of COVID-19, the safety and efficacy of vaccination and new vaccines, disseminating information on security measures taken at vaccination sites, improving confidence in governmental authorities, and understanding individuals’ past experiences with new vaccines are important to reduce COVID-19 vaccine hesitancy. However, using targeted sensitization according to age group or education level may not be as effective. Even though there were challenges in meeting the original target age groups for COVID-19 vaccination during rollout, neither age nor education were found to have an association with vaccine acceptance in our study [7]. Prior similar studies in other countries have also shown inconsistency regarding the association of demographic factors, which are highly context-dependent, with vaccine hesitancy [42,43,44]. 

Despite high acceptance of the Ebola vaccine among this population (83.8% of CMs and 99% of eligible HCWs), the intention to receive the COVID-19 vaccine was still much lower, with vaccine safety and distrust of vaccine manufacturers as major concerns, despite the history of this population with experimental vaccine doses. This may also be due to differences in the perceived risk and severity of the two diseases, as well as high efficacy of the Ebola vaccine [45]. Notably, CMs who had received an Ebola vaccine were more willing to receive the COVID-19 vaccine than other CMs. All respondents in our survey had prior experience with Ebola and were heavily targeted during the Ebola vaccine campaigns. This may have influenced those who received the Ebola vaccine to have more favorable attitudes toward another new vaccine (i.e., COVID-19). A prior survey also found a greater willingness to receive the COVID-19 vaccine among HCWs in Beni and Mbandaka (the regions with Ebola vaccine implementation during an EVD outbreak) compared to those from Kinshasa (a region that has never had an EVD outbreak and EVD vaccination) [40].

The intention to take the COVID-19 vaccine was relatively low (40%) among HCWs in this survey, which is similar to findings from a prior survey of COVID-19 perceptions among HCWs across the DRC from the same time period [40]. Explanations for this finding may be related to the timing of this survey, which occurred during a period of national vaccine delay, as well as heightened concerns about vaccine side effects. Results from a longitudinal cohort survey of HCWs in DRC also pointed out a sharp increase in vaccine hesitancy (i.e., uncertainty or unwillingness to receive a vaccine) in mid-March 2021, followed by a decline by late June 2021, and the most consistent concern was whether the COVID-19 vaccine had been used for a prolonged period without serious side effects [40]. These fluctuations show the dynamic nature of vaccine attitudes and how they may be affected by global influences, especially predominant media narratives. The vaccination of HCWs remains a top priority, given their influential role in the vaccination decisions of their patients and communities, and their role in preventing COVID-19 transmission. Educating HCWs on the safety and side effects of COVID-19 vaccines is also critical because they are the key disseminators of health information [46,47].

Respondents who reported that security concerns prevent access to vaccines and health services were less willing to receive the COVID-19 vaccine. In communities affected by conflict in North Kivu, the benefit of COVID-19 vaccination must be weighed realistically against the security risks. A 2020 qualitative study among internally displaced persons in North Kivu revealed competing concerns with the restoration of peace and security holding greater precedence over COVID-19 vaccination [17]. Additionally, discrepancies in COVID-19 vaccine attitudes among HCWs across the different provinces in DRC may be due to contextual differences in security threats and ongoing conflict in North Kivu. For example, the Equateur province, which has also experienced a concurrent Ebola and COVID-19 outbreaks but did not have the same security issues, has had higher vaccine acceptance [40]. However, the ongoing security threats in Butembo have posed a major barrier to implementing vaccine communication campaigns, resulting in a lower vaccine acceptance rate (unpublished data, SANRU).

HCWs who did not trust the government were less willing to be vaccinated. Perceptions of the governmental capacity to effectively respond to outbreaks and manage healthcare systems are linked to overall trust in authorities. Trust in the government has also been associated with a greater willingness to receive the COVID-19 vaccine among HCWs in diverse settings including Ghana, Ethiopia, and the USA [46,48,49]. Similarly, a DRC household survey evaluating institutional mistrust during the 2018–2020 EVD outbreak found that increased trust in the governments’ EVD response contributed to greater Ebola vaccine acceptance [22]. HCWs in the DRC have worked in extremely challenging conditions (i.e., lacking protective equipment, not being paid consistent salaries which lead to strikes, and workplace violence); this has resulted in strained trust in government authorities [50]. 

HCWs who reported that religion influences “some” (but not “all”) health decisions were less willing to receive the COVID-19 vaccine. Religion plays an important role in the daily lives of the Congolese and religious leaders in the DRC are often trusted and respected community figures who hold significant influence over community attitudes and beliefs, including those toward COVID-19 vaccines [21,51,52]. There have been efforts to include religious leaders in vaccination risk communication and community engagement (RCCE) activities; however, prior research has found that religious leaders sometimes contribute to vaccine concerns by circulating rumors and misinformation [7,53]. Additional research is needed to elucidate the influence of religious leaders and organizations on vaccine perceptions and how to improve engagement and sensitization of religious leaders in COVID-19 vaccine promotion. 

When applied to vaccine hesitancy research, conceptual models, such as the HBM, indicate that vaccine decisions are influenced by a myriad of factors (e.g., perceived susceptibility to the disease and perceived benefits and barriers to vaccination) [33,54,55]. Consistent with the HBM, we found that respondents with a higher perceived risk of contracting COVID-19 were more willing to receive COVID-19 vaccines, which is similar to other reports [56,57]. Notably, only half of the CMs perceived themselves to be at risk of contracting COVID-19, while more than 80% of HCWs felt at risk and showed a higher willingness to receive the COVID-19 vaccine—possibly because of their greater exposure to COVID-19 patients. Our findings contribute to the evidence that low vaccine demand in Africa is likely being driven by the low perceived risk of COVID-19 because the COVID-19 outbreak and the related mortality have been less severe in Africa compared to other regions [57,58,59]. Our survey revealed low levels of willingness to receive COVID-19 vaccines, and this may also be due to the relatively low numbers of documented COVID-19 cases in North Kiv compared to other regions of the DRC, such as Kinshasa, where the vast majority of cases and deaths have occurred [27,40]. Even though the reported numbers of COVID-19 cases and deaths were low in the DRC, the actual number of excess deaths associated with COVID-19 was far higher. The ratio for the excess mortality to reported deaths from COVID-19 was 14:1 for sub-Saharan Africa and 82:1 for the DRC, one of the highest ratios in the world [60]. More accurate reporting of cases and deaths may have potentially decreased vaccine hesitancy.

Based on this survey, males were more willing to receive the COVID-19 vaccine, which is consistent with prior studies [33,42,43,61,62]. Proposed reasons for greater vaccine hesitancy among women include greater concerns regarding safety and side effects, concerns about infertility, and a lower perceived risk of developing severe COVID-19 infection [13,43,63]. The lack of COVID-19 vaccine safety data in pregnant women early in the pandemic, as well as inconsistent and rapidly changing guidelines on vaccine eligibility of pregnant and lactating women, have been found to increase vaccine hesitancy among women [50]. Furthermore, beliefs that the vaccine could cause infertility or have detrimental effects on a developing fetus, have long been women’s concerns in multiple contexts globally [43,64,65]. Gendered approaches to understanding and addressing the specific concerns of women and girls in North Kivu are needed.

Lastly, we found a positive association between general vaccine confidence and the intention to receive the COVID-19 vaccine among both respondent groups. This was consistent with a growing area of research that elucidates how perceptions toward routine vaccinations impact decisions regarding new vaccines. For example, a prior study found that confidence in routine childhood vaccines was a strong predictor of the intention to receive the COVID-19 vaccine among pregnant women and mothers in 16 countries [66,67]. In contrast, a willingness to receive routine vaccines may not consistently predict positive attitudes toward new vaccines deployed in outbreak situations. A community survey in the Western DRC found that the willingness to receive routine vaccinations was high (90%), while the willingness to receive outbreak vaccines (i.e., cholera, Ebola, and COVID-19) was much lower (57%) [68]. In our survey, those who perceived that new vaccines were riskier had a lower intention to receive the COVID-19 vaccine, which supported the evidence of heightened hesitancy toward new vaccines, particularly those introduced during outbreaks [55,56,69]. Although it may not be sufficient to achieve adequate acceptance of new vaccines, increasing general vaccine confidence could also increase confidence in the new vaccines. 

### Limitations

There are several limitations of this study. First, given the timing of the survey in March 2021, the vaccine was not yet available in the DRC; thus, our results only represent the intention to receive a vaccine rather than the actual vaccine acceptance. However, our results are still highly relevant to current vaccination promotion efforts, owing to persistently low vaccine acceptance in this region. Most factors associated with the intention to receive the vaccine (perceived risk of COVID-19, attitudes toward new vaccines, general vaccine confidence, security concerns, religious influence, and trust in government) remain highly amenable to intervention. Second, detailed reasons for vaccine acceptance, delay, and refusal could not be captured with this survey assessment. However, qualitative data were being collected concurrently, and that will greatly contribute to our understanding of socio-behavioral factors influencing COVID-19 vaccination. Third, given the cross-sectional nature of the survey, potential variations in vaccine perceptions over time could not be assessed. Additionally, we did not assess other potential predictors and confounders (e.g., COVID-19 exposures or the presence of chronic medical conditions), which may have influenced perceptions toward COVID-19 vaccines [43]. Lastly, given this unique population, the results of this survey cannot be generalized to other populations globally or even to other populations in the DRC. However, these findings may be valuable to those aiming to improve vaccine confidence during outbreaks among crisis-affected populations and those experiencing multiple concurrent EID outbreaks.

## 5. Conclusions

Although the COVID-19 vaccine rollout began in May 2021 in North Kivu, vaccine acceptance remains low. This survey explored the intention to receive the COVID-19 vaccine and vaccine perceptions before the vaccine’s introduction in North Kivu. Our findings contribute to an improved understanding of the concerns of a unique population affected by previous EID outbreaks and active conflict, and their perceptions toward COVID-19 vaccination. These results could support the success of the current and future vaccine campaigns in this and similar populations. To better control future COVID-19 resurgence in this region, enhancing risk communication efforts and engaging with communities via their preferred means of health communication identified through this survey and others are greatly needed. Community engagement and communication efforts that address the multiple intersecting concerns highlighted by CMs and HCWs would be impactful. We found that both CMs and HCWs showed low willingness to receive the COVID-19 vaccine. The factors associated with the willingness and intent to receive the COVID-19 vaccine were the perceived risk of COVID-19, prior acceptance of the Ebola vaccine, confidence in routine immunizations, concerns about the safety and side effects of new vaccines, security concerns, religious influence, and trust in government. These findings emphasize that addressing security concerns and establishing trust in authorities are critical to improving vaccine confidence in crisis-affected populations. In particular, building trust in public healthcare systems and authorities among North Kivu-based HCWs and involving the government in addressing HCW concerns are of utmost importance when planning vaccine promotion interventions in this priority group. Lastly, considering communities’ prior experiences toward routine and novel vaccines is important when designing interventions to reduce hesitancy toward COVID-19 and other vaccines for EIDs in North Kivu, other humanitarian settings, and populations experiencing concurrent EID outbreaks.

## Figures and Tables

**Table 1 vaccines-11-00973-t001:** Characteristics of community members (CMs) and healthcare workers (HCWs) survey respondents, North Kivu, The Democratic Republic of the Congo, 2021.

Characteristic		CMs (N = 631) n (%)	HCWs (N = 438) n (%)
Age (years), median (IQR) ^1^		31 (22–42)	35 (29–42)
Sex			
Male		208 (33.0)	203 (46.4)
Female		423 (67.0)	235 (53.7)
Health Zone			
Beni		239 (37.9)	167 (38.1)
Butembo		250 (39.6)	172 (39.3)
Mabalako		142 (22.5)	99 (22.6)
Highest Education Level			
None		72 (11.4)	10 (2.3)
Primary school		175 (27.7)	33 (7.5)
Secondary school		324 (51.3)	178 (40.6)
University or higher institute		56 (8.9)	215 (49.1)
Do not know/declined		4 (0.6)	2 (0.5)
Religion			
Catholic		352 (55.8)	227 (51.8)
Protestant/Evangelical/Pentecostal/Revival		250 (39.6)	196 (44.7)
Muslim		17 (2.7)	1 (0.2)
Other ^2^		12 (1.9)	14 (3.2)
Influence of Faith on Decisions Including Health			
No influence		213 (33.8)	152 (34.7)
Influences some decisions		205 (32.5)	170 (38.8)
Influences all decisions		207 (32.8)	114 (26.0)
Declined to respond		6 (1.0)	2 (0.5)
Ebola Vaccination Status			
Received vaccine		397 (62.9)	416 (95.0)
Declined vaccine		77 (12.2)	4 (0.9)
Ineligible or not offered vaccine		157 (24.9)	18 (4.1)
Primary Occupation			
Community Members	n (%)	Healthcare Workers	n (%)
Farmer	181 (28.7)	Nurse	209 (47.7)
Unemployed	113 (17.9)	Doctor	20 (4.6)
Homemaker	93 (14.7)	Administrator	46 (10.5)
Student	68 (10.8)	Hygienist	76 (17.4)
Trader/businessperson	69 (10.9)	Midwife	14 (3.2)
Healthcare worker	24 (3.8)	Lab Technician	25 (5.7)
Work from home	21 (3.3)	Physiotherapist	3 (0.7)
Teacher	9 (1.4)	Medical/Nursing Student	19 (4.3)
Other ^3^	53 (8.4)	Data Manager	10 (2.3)
		Pharmacist	4 (0.9)
		Other ^3^	12 (2.7)

^1^ All figures reported as n (%) except when indicated; unweighted percentages. ^2^ Other religions include: Animist, Atheist, Anglican, and Jehovah’s Witness. ^3^ Other CM occupations include the following (each listed occupation with fewer than five responses): fisherman, traditional healer, seamstress, carpenter, driver, electrician, gardener, engineer, plumber, mason, shoemaker, and military personnel. Other HCW occupations include the following (each listed occupation with fewer than five responses): lab assistant, receptionist, pharmacy worker, and secretary.

**Table 2 vaccines-11-00973-t002:** Knowledge and awareness regarding COVID-19 among community members (CMs) and healthcare workers (HCWs), North Kivu, The Democratic Republic of the Congo, 2021.

Questionnaire Item	CMs (N = 631)		HCWs (N = 438)	
	n	%	n	% (95% CI) ^1^
Have you heard of COVID-19?				
Yes	618	97.9	434	99.1 (98.6, 99.4)
No	7	1.1	1	0.2 (0.0, 0.6)
Unsure/declined	6	1.0	3	0.7 (0.4, 1.2)
Do you think you are at risk of contracting COVID-19?				
Yes	338	53.6	358	81.7 (79.4, 83.9)
No	199	31.5	55	12.6 (10.9, 14.5)
Unsure/declined	94	14.9	25	5.7 (4.6, 7.1)
COVID-19 Transmission: COVID-19 is spread…				
From person-to-person	392	62.1	304	69.4 (66.2, 72.4)
Through coughs and sneezes	403	63.9	336	76.7 (74.1, 79.1)
From animals	66	10.5	46	10.5 (8.6, 12.7)
COVID-19 Prevention: COVID-19 can be prevented by…				
Wearing a mask	540	85.6	420	95.9 (94.6, 96.9)
Washing hands regularly	492	78.0	366	83.6 (81.2, 85.7)
Staying at least 1 m from others	383	60.7	346	79.0 (76.6, 81.2)
Avoiding crowds	178	28.2	183	41.8 (38.6, 45.1)
Going out only when necessary	39	6.2	45	10.3 (8.5, 12.3)

^1^ 95% CI presented for HCWs as survey methods was used for HCW data analysis based on health facility clustering.

**Table 3 vaccines-11-00973-t003:** Intention to receive the COVID-19 vaccine and reasons for accepting or not accepting a COVID-19 vaccine among community members (CMs) and healthcare worker (HCWs) survey respondents, North Kivu, The Democratic Republic of the Congo, 2021.

Questionnaire Item	CMs (N = 631)		HCWs (N = 438)	
	n	%	n	% (95% CI) ^1^
Would you receive a COVID-19 vaccine if offered?				
Yes	174	27.6	174	39.7 (37.0, 42.5)
Unsure	142	22.5	116	26.5 (23.7, 29.5)
No	304	48.2	142	32.4 (29.8, 35.1)
Declined to respond	11	1.7	6	1.4 (9.4, 2.0)
Reasons I would accept a COVID-19 vaccine ^2^	n (N = 316)	%	n (N = 290)	% (95% CI) ^1^
To protect myself and my family	242	76.6	216	74.5 (70.6, 78.0)
To protect other people in my community	166 (52.5)	52.5	138	47.6 (43.4, 51.8)
To stop the spread of COVID-19 in my community	126 (39.9)	39.9	112	38.6 (35.2, 42.2)
Reasons I would not accept a COVID-19 vaccine ^3^	n (N = 304)	%	n (N = 142)	% (95% CI) ^1^
I do not have enough information about the vaccine	161	53.0	94	66.2 (62.0, 70.1)
I am worried that the vaccine is not safe	113	37.2	52	36.6 (32.0, 41.6)
I am worried that the vaccine does not prevent COVID-19	41	13.5	12	8.5 (6.4, 11.1)
I do not trust the local vaccination team	29	9.5	5	3.5 (2.2, 5.6)
I do not trust the manufacturer of the vaccine	82	27.0	39	27.5 (23.3, 32.1)
I do not trust the government	51	16.8	21	14.8 (11.3, 19.2)
Fear of getting COVID-19 while getting vaccinated (e.g., exposure to vaccinators, etc.)	39	12.8	20	14.1 (11.2, 17.6)
Fear of getting COVID-19 from the vaccine itself	39	12.8	16	11.3 (8.9, 14.1)
Would you recommend a COVID-19 vaccine to patients?			n (N = 438)	% (95% CI) ^1^
Yes	n/a		239	54.6 (51.5, 57.6)
Unsure			99	22.6 (20.1, 25.3)
No			90	20.5 (18.3, 23.0)
Declined to respond			10	2.3 (1.6, 3.2)

^1^ 95% CI presented for HCWs as survey methods used for HCW data analysis. ^2^ Among respondents reporting they would take a vaccine or were unsure. Multiple selections are allowed; therefore, proportions do not sum to 100%. ^3^ Among respondents reporting they would not take a vaccine. Multiple selections are allowed; therefore, proportions do not sum to 100%. Abbreviations: CI, confidence interval; n/a, not applicable.

**Table 4 vaccines-11-00973-t004:** Vaccine communication preferences among community members (CMs) and healthcare workers (HCWs), North Kivu, The Democratic Republic of the Congo, 2021.

Questionnaire Item	CMs (N = 631)		HCWs (N = 438)	
	n	%	n	% (95% CI) ^1^
How would you/your community prefer to receive communication on vaccinations and health services in the future? ^2^				
Radio	461	73.1	353	80.6 (78.1, 82.9)
Healthcare workers	356	56.4	313	71.5 (68.5, 72.3)
Religious venues (church, mosque, or other)	311	49.3	271	61.9 (58.5, 65.1)
Megaphone announcements	269	42.6	192	43.8 (40.5, 47.2)
Community leaders (chief or village headman)	157	24.9	162	37.0 (33.7, 40.4)
Other community settings	123	19.5	159	36.3 (33.2, 39.5)
Print materials/flyers	89	14.1	147	33.6 (30.4, 36.9)
Television	104	16.5	111	25.4 (22.2, 28.7)
Outbreak response workers	100	15.9	81	18.5 (16.2, 21.1)
Mobile phone/text message	49	7.8	72	16.4 (14.3, 18.9)
Ministry of health/governmental authority	73	11.6	68	15.5 (11.6, 18.0)
Internet/social media/Facebook/blogs	25	4.0	42	9.6 (7.9, 11.6)
Other	41	6.5	18	4.1 (3.2, 5.3)
What communication methods would you recommend for sharing information on vaccinations to HCW in the future? ^2^				
Meetings/workshops	n/a		375	85.6 (83.3, 87.7)
Posters			227	51.8 (48.8, 54.9)
Pamphlets/handouts			184	42.0 (38.7, 45.3)
Memory aids			108	24.7 (22.0, 27.5)
Mobile phone/text message/Whatsapp			72	16.4 (14.5, 18.6)
Internet/social media/Facebook/Twitter			49	11.2 (9.5, 13.2)
Other			63	14.4 (12.3, 16.8)

^1^ 95% CI presented for HCWs as survey methods used for HCW data analysis. ^2^ Multiple selections are allowed; therefore, proportions do not sum to 100%. Abbreviations: CI, confidence interval. Abbreviations: CI, confidence interval; n/a, not applicable.

**Table 5 vaccines-11-00973-t005:** Correlates of COVID-19 vaccine acceptance and hesitancy among community members (CMs), North Kivu, The Democratic Republic of the Congo, 2021.

	Vaccine Acceptant n (%) N = 174	Vaccine Hesitant n (%) N = 444	RR (95% CI)	aRR (95% CI)
Sex				
Male	68 (39.1)	135 (30.4)	Reference	Reference
Female	106 (60.9)	309 (69.6)	0.77 (0.58–1.03)	0.78 (0.63, 0.96)
Age (years), median (IQR)	31 (22.75, 42.25)	30 (22, 42)	1.00 (0.98, 1.01)	1.00 (0.99, 1.01)
Highest Education Attained				
None	17 (9.8)	55 (12.4)	Reference	Reference
Primary	49 (28.2)	124 (27.9)	1.17 (0.76, 1.80)	1.13 (0.75, 1.72)
Secondary	82 (47.1)	234 (52.7)	1.22 (0.79, 1.89)	1.07 (0.72, 1.60)
University or higher	25 (14.4)	28 (6.3)	2.34 (1.34, 4.08)	1.45 (0.86, 2.44)
Declined to respond	1 (0.6)	3 (0.7)		n/a
Religion Influence				
No influence	66 (37.9)	144 (32.4)	Reference	Reference
Influences some decisions	53 (30.5)	146 (32.9)	0.80 (0.56, 1.15)	0.73 (0.52, 1.04)
Influences all decisions	55 (31.6)	148 (33.3)	0.80 (0.51, 1.24)	0.70 (0.47, 1.06)
Declined to respond	0 (0)	6 (1.4)	n/a	n/a
Perceived Risk of Contracting COVID-19				
No/unsure	44 (25.3)	241 (54.3)	Reference	Reference
Yes	130 (74.7)	203 (45.7)	2.74 (1.92, 3.90)	2.17 (1.54, 3.07)
Vaccine Side Effects Important				
No/unsure	91 (52.3)	249 (56.1)	Reference	Reference
Yes	83 (47.7)	195 (43.9)	1.09 (0.76, 1.58)	0.82 (0.60, 1.12)
Vaccine Efficacy Important				
No/unsure	44 (25.3)	175 (39.4)	Reference	Reference
Yes	130 (74.7)	269 (60.6)	1.62 (1.17, 2.26)	1.13 (0.89, 1.44)
New Vaccines Pose More Risks ^1^				
No/unsure	128 (73.6)	225 (50.7)	Reference	Reference
Yes	46 (26.4)	219 (49.3)	0.46 (0.34, 0.61)	0.63 (0.45, 0.86)
Distrust Government for Vaccine Decisions				
No/unsure	140 (80.5)	274 (61.7)	Reference	Reference
Yes	34 (19.5)	170 (38.3)	0.52 (0.35, 0.77)	0.80 (0.55, 1.15)
Security Concerns Prevent Access to Vaccines and Health Services				
No/unsure	1411 (81.0)	309 (69.6)	Reference	Reference
Yes	33 (19.0)	135 (30.4)	0.69 (0.48, 1.01)	0.70 (0.51, 0.96)
Received Ebola Vaccine				
No	16 (9.2)	59 (13.3)	Reference	Reference
Yes	130 (74.7)	257 (57.9)	1.92 (1.40, 2.62)	1.43 (1.05, 1.94)
Ineligible/not offered	28 (16.1)	128 (28.8)	n/a	n/a
COVID-19 Knowledge Score				
Low	27 (15.5)	78 (17.6)	Reference	Reference
Medium	111 (63.8)	302 (68.0)	1.11 (0.75, 1.64)	0.69 (0.50, 0.96)
High	36 (20.7)	64 (14.4)	1.49 (0.98, 2.28)	0.80 (0.52, 1.23)
General Vaccine Confidence ^2^				
Low	9 (5.2)	72 (16.2)	Reference	Reference
Medium	51 (29.3)	200 (45.1)	1.96 (0.94, 4.08)	2.06 (1.08, 3.90)
High	114 (65.5)	172 (38.7)	3.80 (1.75, 8.26)	3.01 (1.51, 5.99)

Note: 13 Respondents who declined to respond to the question “Would you take a COVID-19 vaccine?” were excluded from regression analysis. Abbreviations: RR: relative risk; aRR: adjusted relative risk. ^1^ Versus older/existing vaccines. ^2^ A composite score was computed using six items. Each question had a scale of 0–3, corresponding to low–high vaccine acceptance; the total composite score was then categorized as low (<6), medium (6–12), or high vaccine (>12).

**Table 6 vaccines-11-00973-t006:** Correlates of COVID-19 vaccine acceptance among healthcare workers, North Kivu, The Democratic Republic of the Congo, 2021.

	Vaccine Acceptant n (%) N = 174	Vaccine Hesitant n (%) N = 258	OR (95% CI)	aOR (95% CI)
Sex				
Male	91 (52.3)	107 (41.5)	Reference	Reference
Female	83 (47.7)	151 (58.5)	0.65 (0.54, 0.77)	0.60 (0.46, 0.80)
Age (years), median (IQR)	35.5 [20–44.25]	35 [29–40.25]	1.01 (1.00, 1.02)	0.98 (0.96, 0.99)
Highest Education Attained				
None	4 (2.3)	6 (2.3)	Reference	Reference
Primary	17 (9.9)	16 (6.2)	1.59 (0.79, 3.20)	1.43 (0.35, 4.75)
Secondary	67 (38.5)	108 (41.9)	0.93 (0.49, 1.77)	0.47 (0.13, 1.62)
University or higher	84 (48.3)	128 (49.6)	0.98 (0.51, 1.90)	0.37 (0.11, 1.22)
Religion Influence	0 (0)	2 (1.2)		
No influence	68 (39.1)	81 (31.6)	Reference	Reference
Influences some decisions	55 (31.6)	115 (44.9)	0.57 (0.44, 0.73)	0.45 (0.34, 0.61)
Influences all decisions	51 (29.3)	60 (23.4)	1.01 (0.78, 1.31)	0.93 (0.68, 1.27)
Declined to respond	2 (0.8)	0 (0)	n/a	n/a
Perceived Risk of Contracting COVID-19				
No/unsure	22 (12.6)	54 (20.9)	Reference	Reference
Yes	152 (87.4)	204 (79.1)	1.77 (1.33, 2.36)	2.23 (1.53, 3.26)
Vaccine Side Effects Important				
No/unsure	98 (56.3)	127 (49.2)	Reference	Reference
Yes	76 (43.7)	131 (50.8)	1.38 (1.11, 1.72)	0.72 (0.57, 0.91)
Vaccine Efficacy Important				
No/unsure	41 (23.6)	77 (29.8)	Reference	Reference
Yes	133 (76.4)	181 (70.2)	1.38 (1.11, 1.72)	1.30 (0.95, 1.76)
New Vaccines Pose More Risks ^1^				
No/unsure	148 (85.1)	147 (57.0)	Reference	Reference
Yes	26 (14.9)	111 (43.0)	0.17 (0.13, 0.21)	0.23 (0.17, 0.33)
Distrust Government for Vaccine Decisions				
No/unsure	146 (83.9)	171 (66.3)	Reference	Reference
Yes	28 (16.1)	87 (33.7)	0.36 (0.28, 0.46)	0.50 (0.35, 0.70)
Security Concerns Prevent Access to Vaccines and Health Services				
No/unsure	141 (81.0)	170 (65.9)	Reference	Reference
Yes	33 (19.0)	88 (34.1)	0.45 (0.36, 0.57)	0.52 (0.37, 0.74)
COVID-19 Knowledge Score				
Low	7 (4.0)	26 (10.1)	Reference	Reference
Medium	107 (61.5)	173 (67.1)	2.30 (1.59, 3.31)	1.37 (0.87, 2.16)
High	60 (34.5)	59 (22.9)	3.78 (2.55, 5.60)	1.62 (0.97, 2.72)
General Vaccine Confidence ^2^				
Low	4 (2.3)	22 (8.5)	Reference	Reference
Medium	36 (20.7)	102 (39.5)	1.94 (1.14, 3.31)	3.73 (1.98, 7.01)
High	134 (77.0)	134 (51.9)	5.5 (3.30, 9.18)	7.16 (3.89, 3.17)

Note: 6 Respondents who declined to respond to the question “Would you take a COVID-19 vaccine?” were excluded from regression analysis. OR: odds ratio; aOR: adjusted odds ratio. ^1^ Versus older/existing vaccines. ^2^ A composite score was computed using six items. Each question had a scale of 0–3, corresponding to low–high vaccine acceptance; the total composite score was then categorized as low (<6), medium (6–12), or high vaccine (>12).

## Data Availability

The de-identified dataset is available upon reasonable request to the corresponding author.

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
