# Peer review of "COVID-19 Vaccine Perceptions among Ebola-Affected Communities in North Kivu, Democratic Republic of the Congo, 2021"

_vaccines, 2023, doi:10.3390/vaccines11050973_

Round 1
Reviewer 1 Report
There is no doubt that the awareness investigation of the COVID19 vaccine in the DRC, which has been put in various conflict situation, must be recognized by everyone in terms of humanity. However, the blind point is that it seems that people in other countries would have answered that way before the introduction of the vaccine.
I would like to recommend that this manuscript (MS) should be published out of a sense of humanitarian responsibility to inform the situation in the DRC. However, the MS is too long and verbose, and I doubt whether it is necessary to include almost all questionnaire in the form of a table.
Discussion is too long to be a published paper. Cases of COVID19 vaccines not being used and expired due to vaccine refusal can be found all over the world, i.d., this is not only for the DRC. It is uncomfortable to read what is described as being driven by this country alone.
The introduction and discussion should be re-written concisely, and numerous tables filled with results also need to be sorted out somehow.
Author Response
There is no doubt that the awareness investigation of the COVID19 vaccine in the DRC, which has been put in various conflict situation, must be recognized by everyone in terms of humanity. However, the blind point is that it seems that people in other countries would have answered that way before the introduction of the vaccine.
Cases of COVID19 vaccines not being used and expired due to vaccine refusal can be found all over the world, i.d., this is not only for the DRC. It is uncomfortable to read what is described as being driven by this country alone.
Response: Thank you very much for these constructive comments. The reviewer is correct that extensive research has been conducted globally on COVID-19 vaccine hesitancy, with common factors found across diverse contexts. However, we emphasize that there is a significant lack of information on populations in humanitarian contexts and in North Kivu, specifically. We clarify that we do not suggest that these factors are exclusive to or being driven by the DRC, but that we have sought to understand the similarities and differences towards COVID-19 vaccine perceptions among this unique population compared to other populations in DRC as well as other similar populations.
We have clarified that the results from this population cannot necessarily be generalized to other populations globally, or even other populations in the DRC, although the findings may be particularly useful for populations affected by humanitarian crises, and those experiencing multiple concurrent epidemics of emerging infectious diseases. We have added these points to the Limitations section, “Lastly, given this unique population, the results of this survey cannot be generalized to other populations globally, or even to other populations in the DRC. However, these findings may be valuable to those aiming to improve vaccine confidence for EIDs in crisis-affected populations and those experiencing multiple or concurrent EID outbreaks.”
I would like to recommend that this manuscript (MS) should be published out of a sense of humanitarian responsibility to inform the situation in the DRC. However, the MS is too long and verbose, and I doubt whether it is necessary to include almost all questionnaire in the form of a table. Discussion is too long to be a published paper.
The introduction and discussion should be re-written concisely, and numerous tables filled with results also need to be sorted out somehow.
Response: We have condensed the Introduction, Results and Discussion Sections. However, please note that because Reviewer 2 conversely requested that the Discussion section be significantly expanded, we have tried to balance the two reviewer requests as much as possible, including moving parts of the discussion into the conclusion section as requested by the editor. We have also moved Table 3 to supplementary materials. We hope this satisfies the reviewer concerns and has improved the manuscript.
Reviewer 2 Report
The manuscript entitled "COVID-19 Vaccine Perceptions among Ebola-affected Communities in North Kivu, Democratic Republic of the Congo, 2021" Title, abstract and overall rationale of work is well written. However, there are still some minor concerns, which needs to be addressed before publication
1) The keywords should be revised and do not repeat the title word For instance, SARS-CoV-2 and COVID-19 refers to the same.
2) Introduction section: Author should be re-write introduction section in concise way and author also describe which vaccine available/taken in this country. Moreover, what is the availability of COVID-19 vaccine in this country?
3) Author also need to explain about the Ebola virus disease and their status in this country.
4) In the material method section: Author must be write the Ebola virus vaccine name and company in the method section. Author should be add survey methodology and diagram to show all these data.
5) Results section: This section is written clearly and explain in deep.
6) Author need expend discussion and much more explanations and interpretations must be added for the results, which are not enough at all. It is suggested to compare the results of the present research with some similar studies which is done before.
7) Conclusion section must be elaborate and this section should present at least in one 250-300 words paragraph and author must write future prospective and significance of this study.
English Quality of this manuscript is good.
Author Response
The manuscript entitled "COVID-19 Vaccine Perceptions among Ebola-affected Communities in North Kivu, Democratic Republic of the Congo, 2021" Title, abstract and overall rationale of work is well written. However, there are still some minor concerns, which needs to be addressed before publication
1) The keywords should be revised and do not repeat the title word For instance, SARS-CoV-2 and COVID-19 refers to the same.
Response: Thank you very much for this observation, we have removed COVID-19 from key words as it is already included in the title.
2) Introduction section: Author should be re-write introduction section in concise way and author also describe which vaccine available/taken in this country. Moreover, what is the availability of COVID-19 vaccine in this country?
Response: Thank you very much for this comment, we have shortened the Introduction section and clarified which vaccines were available during vaccine roll-out when the survey was conducted, “Vaccinations in the DRC began with roll-out of the ChAdOx1-S [recombinant] (AstraZeneca®/Covishield) vaccine in April 2021 and subsequently in September 2021 the Moderna mRNA-1273 and Pfizer-BioNTech (BNT162b2) COVID-19 vaccines , through the COVID-19 Vaccines Global Access (COVAX) facility, and the Sinovac vaccine through a bilateral agreement between DRC and China.”
3) Author also need to explain about the Ebola virus disease and their status in this country.
Response: Thank you very much for this comment, we clarify that we have focused the introduction on the 2018-2020 Ebola outbreak in DRC given the parent survey was conducted to understand perceptions toward Ebola vaccination during that outbreak. There are currently no outbreaks of Ebola in DRC, and the most recent Ebola outbreak in DRC was declared over in September 2022, 42 days after the death of the only confirmed case. We have referenced this most recent outbreak in the introduction section, “North Kivu, a province located in the Eastern DRC, has been the site of decades-long humanitarian emergencies and an active conflict zone. It has also been the site of multiple outbreaks of EVD (including the most recent in August 2022), and was at the epicenter of the 2018–2020 EVD outbreak, the second largest in history, resulting in over 3,400 cases and 2,200 deaths [12].”
4) In the material method section: Author must be write the Ebola virus vaccine name and company in the method section. Author should be add survey methodology and diagram to show all these data.
Response: Thank you very much for this comment, we have added the name of the Ebola vaccine (rVSV-ZEBOV) and manufacturer Merck to methods section as suggested. We have added a flow diagram to supplemental figures as suggested.
5) Results section: This section is written clearly and explain in deep.
Response: Thank you very much!
6) Author need expend discussion and much more explanations and interpretations must be added for the results, which are not enough at all. It is suggested to compare the results of the present research with some similar studies which is done before.
Response: Thank you for these comments, we have clarified where comparisons have been made between our findings and similar studies especially those in DRC or similar contexts, and restructured our discussion and conclusion sections to focus on implications of the findings. However, please note that since Reviewer 1 conversely requested that the Discussion section be significantly shortened, we have tried to balance the two reviewer requests as much as possible, keeping the Discussion section as concise as possible.
7) Conclusion section must be elaborate and this section should present at least in one 250-300 words paragraph and author must write future prospective and significance of this study.
Response: Thank you very much for this comment, we have expanded the Conclusion section including focusing on recommendations based on this survey’s findings and adding broader implications to other crisis-affected populations and those experiencing multiple emerging infectious disease outbreaks.
Reviewer 3 Report
In this study, the acceptance rate of the Covid-19 vaccine in the among population affected by the Ebola Virus Disease outbreak in North Kivu, Democratic 27 Republic of the Congo has been investigated. The effect of many demographic, social, vaccine attitude and etc. factors on this rate has been studied in two groups: community members and healthcare workers. The methodology is explained well and clearly and in detail. This study will be very helpful for regional and national health policy makers. However, following comments should be addressed in the next revisions.
Abstract
-In the findings section of the abstract, where the relationship between the variables and vaccine perceptions is expressed, the coefficients/odds ratios and their significance should be reported.
Methods
For the dependent variable of this study, logistic or probit models are usually suggested. Also, the Poisson model is used when the dependent variable is a count variable, such as the number of doctor visits. It is not clear to us why Poisson model was used in this study. Explain more about the selection of regression model.
Results
In the table 6 and 7, provide measures of regression models goodness of fit.
Discussion
-While your study offers valuable insights into the unique characteristics and circumstances of the study area, it may not be possible to generalize the findings to other areas or populations with distinctive socio-economic and underlying circumstances of this area. Therefore, it is important to acknowledge the limitations of the study and the implications of the results in light of the study area's unique conditions. Moreover, other factors neglected unintendedly as it is stated in limitations of the current paper, such as clinical concerns for the side effects of the COVID vaccines as claimed significantly had its own effect on the Vaccine Perceptions.
- The global implications of the study's results necessitate further discussion. It is crucial to explore the potential impact of the findings on a broader scale. The study warrants a broader consideration of its implications beyond its immediate context.
Author Response
In this study, the acceptance rate of the Covid-19 vaccine in the among population affected by the Ebola Virus Disease outbreak in North Kivu, Democratic 27 Republic of the Congo has been investigated. The effect of many demographic, social, vaccine attitude and etc. factors on this rate has been studied in two groups: community members and healthcare workers. The methodology is explained well and clearly and in detail. This study will be very helpful for regional and national health policy makers. However, following comments should be addressed in the next revisions.
Abstract
-In the findings section of the abstract, where the relationship between the variables and vaccine perceptions is expressed, the coefficients/odds ratios and their significance should be reported.
Response: Due to suggested abstract word limits, these figures were not originally included. We have now added the appropriate measures of effect with 95% confidence intervals to the findings section of the abstract as suggested.
Methods
For the dependent variable of this study, logistic or probit models are usually suggested. Also, the Poisson model is used when the dependent variable is a count variable, such as the number of doctor visits. It is not clear to us why Poisson model was used in this study. Explain more about the selection of regression model.
Response: Thank you very much for this comment, we have clarified in methods that the modified Poisson model as initially described by Zou et al (2004) is an alternative to binary logistic regression, when the outcome is clustered and not rare which would result in overestimation of risk. In contrast, the standard Poisson regression model is used for count outcomes as the reviewer notes.
Results
In the table 6 and 7, provide measures of regression models goodness of fit.
Response: Thank you for this request, we have added measures of goodness-of-fit for the regression models to the appropriate section of methods and results.
Methods: “Goodness-of-fit was assessed for the regression models using the F-adjusted mean residual test for the HCW data and chi‐square goodness-of-fit test for the community members data.”
Results: For HCWs – “The model’s F-adjusted mean residual test p-value was 0.31 indicating no evidence of lack of fit.” For community – “The model’s chi-square goodness-of-fit test p-value was 0.18 indicating no evidence of lack of fit.”
Discussion
-While your study offers valuable insights into the unique characteristics and circumstances of the study area, it may not be possible to generalize the findings to other areas or populations with distinctive socio-economic and underlying circumstances of this area. Therefore, it is important to acknowledge the limitations of the study and the implications of the results in light of the study area's unique conditions. Moreover, other factors neglected unintendedly as it is stated in limitations of the current paper, such as clinical concerns for the side effects of the COVID vaccines as claimed significantly had its own effect on the Vaccine Perceptions.
Response: Thank you for these observations, we concur that the results from this unique population cannot necessarily be generalized to other populations globally, or even other populations in the DRC, although the findings may be particularly useful for populations affected by humanitarian crises, and those experiencing multiple concurrent epidemics of emerging infectious diseases. We have added these points to the Limitations section, “Lastly, given this unique population, the results of this survey cannot be generalized to other populations globally, or even to other populations in the DRC. However, these findings may be valuable to those aiming to improve vaccine confidence for EIDs in crisis-affected populations and those experiencing multiple concurrent EID outbreaks.”
- The global implications of the study's results necessitate further discussion. It is crucial to explore the potential impact of the findings on a broader scale. The study warrants a broader consideration of its implications beyond its immediate context.
Response: Thank you for this comment, as mentioned above, given the unique population, while it would be inappropriate to generalize these findings to global populations broadly, we have added what populations these findings would be particularly valuable for, namely those experiencing humanitarian crises, as well as concurrent outbreaks of emerging infectious diseases that may be curbed by vaccination. In particular, the findings highlight the importance of addressing security concerns and vaccine confidence towards routine and outbreak vaccines over the long-term in order to ensure adequate vaccine uptake in future outbreaks. These broader implications and recommendations for those planning interventions to improve vaccine confidence have been added to the Conclusion section.
Round 2
Reviewer 1 Report
Responses to reviewer's comment was fine enough to publish.
Reviewer 2 Report
The authors have addressed all the concerns raised in the previous version of the manuscript and the quality has much improved after incorporating required modifications. Therefore, the manuscript may be considered for publication in this Journal.